



# The Interactions Between Precipitation, Vegetation and Dust Emission Over Semi-Arid Mongolia

Yuki Sofue[1], Buho Hoshino[2*], Yuta Demura[3], Eunice Nduati[1], Akihiko Kondoh[1]

[1]Department of Earth Sciences, Chiba University, Chiba, 2638522, Japan
[2]Laboratory of Environmental Remote Sensing, Rakuno Gakuen University, Ebetsu, 0698501, Japan
[3] Field Researchers Corporation CO. Ltd, Sapporo, Hokkaido, 060-0007, Japan

*Correspondence to: Buho Hoshino (aosier@rakuno.ac.jp)

**Abstract.** Recently, droughts have become widespread in the Northern Hemisphere, including in Mongolia. The ground surface condition, particularly vegetation coverage affects the occurrence of dust storms. The main sources of dust storms in the Asian region are Taklimakan and Gobi deserts. The purpose of this study is to examine the relationship between the trend of vegetation variation and the effects of precipitation in the Gobi region. In the Gobi region, precipitation is confined to the period from May to September. We compared the patterns of interactions between precipitation and normalized difference vegetation index (NDVI) for a period of 29 years. The precipitation and vegetation datasets were examined to investigate the trends between 1985 - 2013. Cross correlation analysis between the precipitation and the NDVI anomalies was performed. Data analysis showed a decreasing trend in precipitation amount and its spatial shift from the east to west part of the region investigated. The vegetation in the area with the lowest precipitation was more sensitive to the precipitation dynamics than those parts with relatively higher values. The most degraded area was the southwest region of Gobi with the least precipitation.

## 1 Introduction

The Gobi includes a great desert and semi-arid region of Central Asia and stretches across huge portions of both Mongolia and China. The characteristic vegetation constitutes mixtures of grasslands, shrubs, saltwort and thorny trees. The Mongolian Gobi is a source area for the formation of dust storms in East Asia (Natsagdorj et al., 2003). Dust storms frequently occur in the arid and semi-arid regions and may have contributed to the desertification, observed in recent decades as well as accelerated occurrence of more arid conditions over the drylands of Asia (Huang et al., 2014). Vegetation coverage is one of the most important factors for the reduction of dust storm occurrence (Ishizuka et al., 2005), and water is the main limiting factor for vegetation growth over south Mongolia (Liu et al., 2013). Precipitation is one of the most important water sources for growth of plants in arid and semi-arid regions. However, both observation and modelling studies have indicated that an aridity trend is occurring and will occur most significantly in the semi-arid regions in terms of precipitation, soil moisture, and drought frequency as a result of global warming (Fu et al., 1999). Furthermore, Huang et al. (2016) point out that the warming trends over drylands, particularly in arid regions, are twice as great as those over humid regions. Sparsely vegetated



drylands are an important source of dust emission, but the mechanism of dust generation in response to timing of precipitation and the consequent effects on soil and vegetation dynamics in these settings is little known (Urban et al., 2009). In this study, we used time series satellite vegetation measurements from National Oceanic and Atmospheric Administration (NOAA) Advanced Very High Resolution Radiometer (AVHRR) sensor to examine the variability and trends of land surface conditions in the Gobi region as represented by vegetation index data for 29 years from 1985 to 2013.

## 2 Data

### 2.1 NDVI data

In this study, we use the Normalized Difference Vegetation Index (NDVI) to estimate vegetation variation. NDVI is given by

$$\text{NDVI} = \frac{NIR - RED}{NIR + RED} \qquad (1)$$

Where, RED and NIR are the surface reflectance bands in the 550–700 nm (visible) and 730–1000 nm (infrared) regions of the electromagnetic spectrum, respectively. The NDVI3g data set used in this study is derived from measurements made by the AVHRR sensor aboard NOAA polar orbiting satellite series (NOAA-7, 9, 11, 14, 16). The NDVI3g data set is provided by the GIMMS group at NASA's Goddard Space Flight Center, as described by Tucker et al. (2005) and cover the period from 1981 to 2013, with a spatial resolution of 8km by 8km. The NDVI data were generated from processed 15-day NDVI composites using the maximum value compositing procedure to minimize effects of cloud contamination, varying solar zenith angles and surface topography (Holben, 1986). For this study, we subset the Gobi region covering the domain $90^0$ E - $117.5^0$ E and $40^0$ N - $47.5^0$ N, from the continental data set for the period January 1985 to December 2013.

### 2.2 Precipitation data

The Global Precipitation Climatology Project (GPCP) data derived from a joint analysis of satellite data and gauge data (Huffman et al., 2009) was used as precipitation data. This data has daily and monthly data. Daily data has $1^0 \times 1^0$ spatial resolution acquired between October 1996 and May 2015. And monthly data has $2.5^0 \times 2.5^0$ spatial resolution acquired between January 1979 and May 2015. Since, the evolution of NDVI in the Gobi region focused on rainfall season, NDVI patterns during the Growing Season (GS) were mainly analyzed.

### 2.3 Dust storm data

Data of NAMHEM (National Agency for Meteorology Hydrology and Environmental Monitoring, Mongolia) was used as the number of days of occurrence of dust storm events. NAMHEM is joined to the World Meteorological Organization (WMO) and has 130 weather stations in the country. The data used is observed at Sainshand weather station (Station ID Number: 443540).



## 3 Method

We examined the spatiotemporal and seasonal variations, as well as the anomaly patterns for the monthly time series from 1985 to 2013. The growing season was defined by examining the long-term mean patterns of both precipitation and NDVI as shown in Fig. 1a and b, respectively, and with reference to long-term patterns of annual average precipitation distribution

(Anyamba and Tucker, 2005). The months of May through September were selected to represent the average start and end of the Growing Season, referred to here as GS. Fig. 1 shows the map of the average of all data for the study period from 1985 to 2013. This shows the long-term mean for this region. Interannual variability in the NDVI pattern was examined by calculating yearly GS anomalies as follows;

$$NDVI\sigma = \left[ \left( (NDVI\alpha)/(NDVI\mu) - 1 \right) \times 100 \right]$$

10  (2)

Where $NDVI\sigma$ are the respective GS percent anomalies, $NDVI\alpha$ are individual seasonal GS means and $NDVI\mu$ is the long-term GS mean. We also examined the precipitation anomalies during GS using same method as that used for NDVI anomalies. Then we used the cumulative values of precipitation during GS. In addition, we performed cross correlation analysis between the cumulative precipitation and averaged NDVI for 15 days by averaging pixels in (1°×1°) to verify the

results of comparison with both trends. In this analysis, we use daily precipitation data and NDVI data resampled to (1°×1°). And analysis period is determined by period of daily precipitation data from 1996 to 2013.

## 4 Results

### 4.1 Spatial patterns and trends

The anomaly time series for the region are represented by the Hovmoller diagram for the period from January 1985 to

December 2013 (Fig. 2a, 2b). It was considered that vegetation trend variation arises due to a difference in conventional precipitation in the monsoon season. Amount of precipitation, which is supplied by monsoon from Pacific Ocean and Indian Ocean, differs greatly between the east and west. Vegetation in the eastern region (from 110° E to 117.5° E) which had higher conventional precipitation had a high response to the precipitation. In 1998 and 2012, it showed a higher than normal response to precipitation. In the eastern region, low amount of precipitation had been retained during 1999 to 2011.

Nevertheless, vegetation anomaly was around 0 or more than 0. In contrast, in the central part of the study area ($100^0$ E from $110^0$ E), high response of vegetation to higher precipitation was observed only three years (1994, 1995, 2003). Also, there was no response to precipitation in 2012. The western region beginning from $100^0$ E to $90^0$ E showed a low response of vegetation to precipitation as compared to the east and central parts starting from 1985 due to the lower amount of conventional precipitation. Fluctuations in precipitation anomaly in the west part became very large since 2000. This is

considered to be due to the recent climate change and that the difference in the conventional precipitation determines the degree of influence. Also, it was found that the vegetation in the western part of the area investigated is more vulnerable to



climate change. The vegetation of the western region had a strong negative trend since 2010 and did not recover in the following years with greater values of precipitation. It is assumed that this decreasing trend might have promoted the reduction of vegetation further.

Time series of NDVI for selected locations across the Gobi region for the period 1985–2013 are shown in Fig. 4. The data

presented here are averaged NDVI values and cumulative precipitation for GS at each point. Sites of 1 and 2 showed no change in trends of NDVI through the time series. On the other hand, site 3 showed a positive trend from 2003, and site 4 showed a negative trend from around 2009. There was a big difference in standard NDVI value sites 1 for 4. Sites 1 and 2 had a relatively large variation of NDVI year to year. Conversely sites 3 and 4 had a small variation. These variations depended on their response to precipitation.

### 4.2 Cross correlation analysis

The results of the cross-correlation analysis across the Gobi region for the period 1996 - 2013 are shown in Fig. 5 and the data is significant at $p < 0.05$ levels. The distribution of correlation coefficient is shown in Table 1. In the eastern region, there was a relatively high trend of correlation coefficient. By contrast, the time lag was larger and correlation coefficient

was very low in the western region, especially in the southwest part. The time lag was almost 0 and the vegetation conditions response within 15 days after precipitation in other regions. Positive relationship between NDVI and precipitation during GS in Sainshand is shown as an example (Fig. 6). The highest correlation value was 0.4 ($R^2 = 0.17$, $p < 0.05$) at time lag 0 locations. The vegetation had decreasing trends, but we postulate that it would recover in most locations during seasons with sufficient precipitation.

**5 Conclusions**

Satellite measurements of vegetation dynamics in the Gobi region for a period of 29 years showed interannual variation and trends. In the Gobi region, precipitation is confined to the period from May to September. The variations of NDVI anomalies in the east region correspond well with the documented precipitation anomalies during this period. However, some parts, especially those in southwest region of Gobi region showed that the NDVI had decreased regardless of precipitation amount.

Semi-arid Gobi region has two kinds of plants: annual plant and perennial plants. Annual plants are highly dependent on rainfall and are also susceptible to overgrazing, while perennial plants are relatively stable and can grow even in extreme drought conditions. However, once perennial plants, e.g. shrubs die, they need a very long time to recover. This is a contributory factor to the occurrence of desertification. Desertification can increase dust storms as has been observed in the Tibet Plateau and Hexi Corridor in recent years. This area is located in Northwest China, including Tarim Basin. Perennial

plants are dominant in this area due to low precipitation and desertification is therefore more likely to occur. Dust storms in



China are caused, to a large degree, by human activities. The western Tarim Basin is one of the areas that has high frequency of dust storm events (Wang et al., 2004).

This study focuses on dynamic interaction between precipitation, vegetation (NDVI) and dust emission, however, only in growth season (GS), the annual grasses are reflected in NDVI. They exist as dead grass in the spring, but are not reflected in

NDVI. Rainfall encourages the growth of annually herbaceous plants and is recorded as a memory of biomass (Dry Matter Productivity), and in the following year they suppress dust emission as dry grass. The differences in dead grass coverage rates may increase or decrease the outbreak of dust storms. The quantity of dust storm emissions tended to decrease along with an increased rate of the dead grass coverage areas with a maximum wind speed exceeded 9.1 m・s-1 in our study sites (Demura, et al., 2016). Similarly, there were some cases where the quantity of dust storm emission had increased when the

dead grass coverage areas had decreased rate at the same maximum wind speed exceeded 9.1 m・s-1. In particular cases, the number of dust storm emissions had a predilection to decrease along with an increased rate of the dead grass coverage even the maximum wind speed exceeded 11 m・s-1 (Demura, et al., 2016).

The relationship between summer vegetation and number of days in which dust storms occurred in the next spring in Sainshand city shown in Fig. 7a, b. These data were analyzed using single regression analysis and the correlation coefficient

was significantly negative, -0.5 ($R^2$ = 0.22, $p$ <0.05, N = 28). Therefore, the conditions of vegetation coverage during GS might influence dust storm frequency. It has been suggested that maintaining vegetation coverage during this period could reduce dust storm occurrence in the next spring. Fig. 5 indicates that the vegetation condition in southwest region of Gobi should be monitored more carefully in the future.

**Acknowledgements**. This work was supported by JSPS KAKENHI Grant Numbers 16H02703, JP24340111, JP25550079 and the joint research of Rakuno Gakuen University. We are grateful to P. Tsedendamba and D. Munkhjargal for their support on field survey in Mongolia.



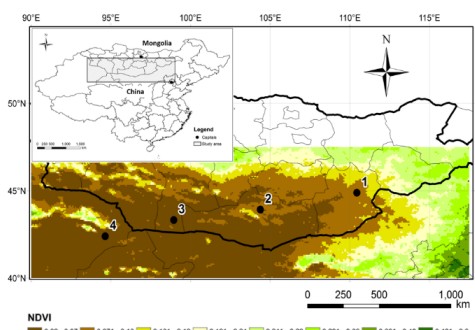

Figure 1: Long-term mean NDVI for the Gobi region (1985–2013) showing the transition from east region with NDVI values of 0.6 to west with values 0.02. The numbered locations indicate sites where NDVI data were extracted to examine the temporal variations and trends in NDVI from 1985 to 2013. The site 1 is located near Sainshand city, the capital of Dornogovi Province in Mongolia.

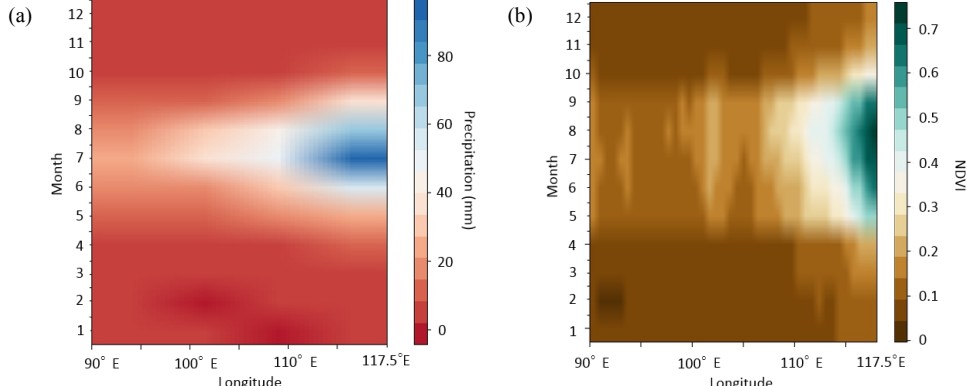

Figure 2: Hovmoller diagrams (a) monthly precipitation and (b) monthly NDVI for the Gobi region averaged between $90^0$ E-$117.5^0$ E and $40^0$ N-$47.5^0$ N.





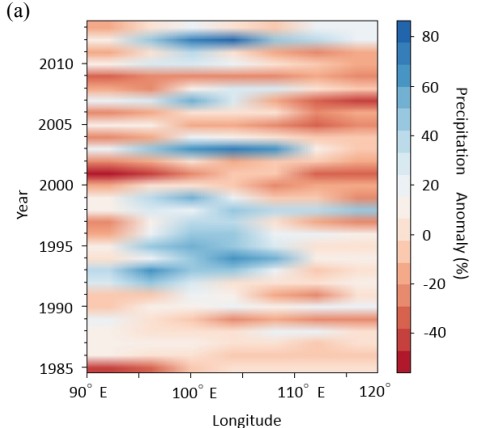

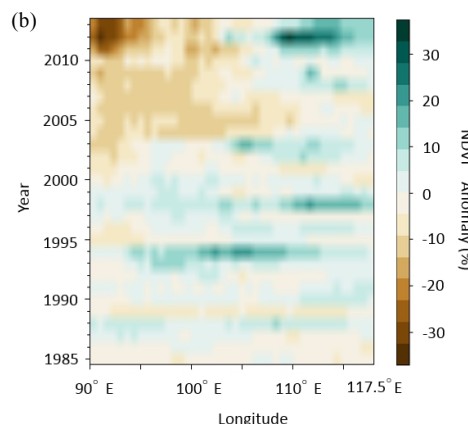

Figure 3 Hovmoller diagrams (a) precipitation anomaly and (b) NDVI anomaly for the Gobi region averaged between $90^0$ E-$117.5^0$ E and $40^0$ N-$47.5^0$N.

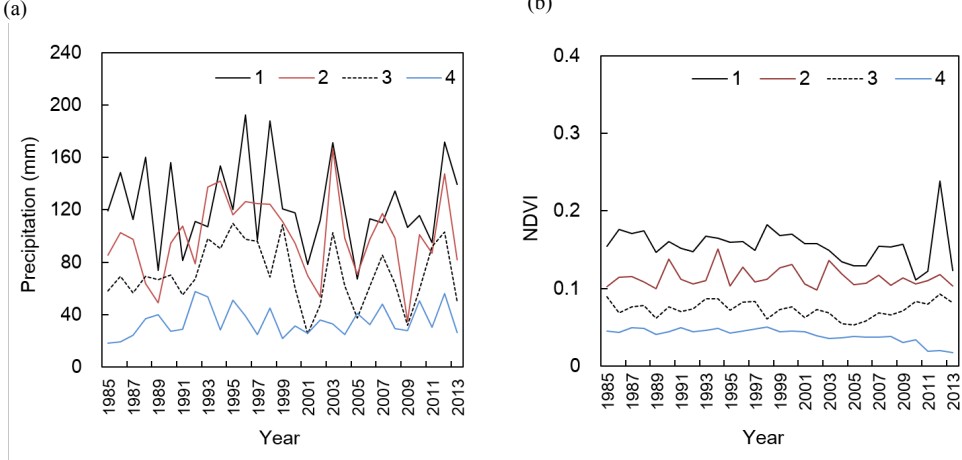

10 Figure 4 Time series of average GS NDVI for selected sites across the Gobi region and cumulative GS precipitation during same period (a, b). All sites, excluding site 4, grow a similar trend of precipitation variation over time. (a). Site 4 shows minimal variation. On the one hand, the distribution patterns of NDVI have indicated decreased amount and shifted location from Northeast to Southwest region (b).

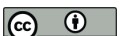


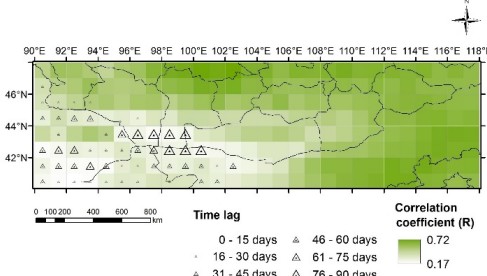

Figure 5 Map of the correlation between precipitation and NDVI, and distribution of time lag in response to precipitation.

Table 1 Correlation coefficient matrix.

| | 90°E | 91°E | 92°E | 93°E | 94°E | 95°E | 96°E | 97°E | 98°E | 99°E | 100°E | 101°E | 102°E | 103°E | 104°E | 105°E | 106°E | 107°E | 108°E | 109°E | 110°E | 111°E | 112°E | 113°E | 114°E | 115°E | 116°E | 117°E |
|---|---|---|---|---|---|---|---|---|---|---|---|---|---|---|---|---|---|---|---|---|---|---|---|---|---|---|---|---|
| 48°N | 0.47 | 0.56 | 0.55 | 0.51 | 0.43 | 0.53 | 0.56 | 0.65 | 0.67 | 0.68 | 0.71 | 0.70 | 0.72 | 0.66 | 0.66 | 0.61 | 0.65 | 0.65 | 0.66 | 0.66 | 0.63 | 0.65 | 0.65 | 0.64 | 0.66 | 0.61 | 0.64 | 0.61 |
| 47°N | 0.38 | 0.48 | 0.51 | 0.56 | 0.49 | 0.46 | 0.50 | 0.56 | 0.56 | 0.59 | 0.65 | 0.67 | 0.66 | 0.61 | 0.60 | 0.57 | 0.57 | 0.56 | 0.60 | 0.61 | 0.59 | 0.58 | 0.60 | 0.64 | 0.61 | 0.60 | 0.60 | 0.60 |
| 46°N | 0.29 | 0.34 | 0.40 | 0.48 | 0.48 | 0.43 | 0.47 | 0.50 | 0.44 | 0.48 | 0.52 | 0.58 | 0.50 | 0.46 | 0.50 | 0.51 | 0.47 | 0.45 | 0.46 | 0.48 | 0.55 | 0.58 | 0.62 | 0.63 | 0.64 | 0.63 | 0.60 | 0.61 |
| 45°N | 0.26 | 0.32 | 0.21 | 0.24 | 0.21 | 0.22 | 0.20 | 0.24 | 0.22 | 0.34 | 0.32 | 0.30 | 0.37 | 0.28 | 0.33 | 0.33 | 0.40 | 0.41 | 0.41 | 0.38 | 0.51 | 0.58 | 0.62 | 0.61 | 0.65 | 0.66 | 0.62 | 0.64 |
| 44°N | 0.37 | 0.24 | 0.40 | 0.51 | 0.43 | 0.19 | 0.17 | 0.18 | 0.17 | 0.16 | 0.27 | 0.28 | 0.34 | 0.42 | 0.43 | 0.44 | 0.40 | 0.36 | 0.42 | 0.46 | 0.52 | 0.56 | 0.60 | 0.61 | 0.66 | 0.68 | 0.68 | 0.68 |
| 43°N | 0.18 | 0.20 | 0.26 | 0.23 | 0.24 | 0.25 | 0.23 | 0.19 | 0.19 | 0.22 | 0.19 | 0.18 | 0.19 | 0.25 | 0.32 | 0.34 | 0.31 | 0.43 | 0.54 | 0.55 | 0.55 | 0.58 | 0.62 | 0.62 | 0.66 | 0.68 | 0.67 | 0.67 |
| 42°N | 0.17 | 0.17 | 0.18 | 0.19 | 0.15 | 0.23 | 0.25 | 0.23 | 0.19 | 0.19 | 0.23 | 0.19 | 0.21 | 0.20 | 0.24 | 0.31 | 0.48 | 0.56 | 0.62 | 0.63 | 0.65 | 0.65 | 0.69 | 0.67 | 0.69 | 0.67 | 0.65 | 0.66 |
| 41°N | 0.17 | 0.17 | 0.17 | 0.17 | 0.24 | 0.28 | 0.26 | 0.35 | 0.30 | 0.27 | 0.25 | 0.25 | 0.24 | 0.29 | 0.34 | 0.40 | 0.49 | 0.54 | 0.59 | 0.63 | 0.65 | 0.67 | 0.69 | 0.69 | 0.65 | 0.64 | 0.59 | 0.60 |

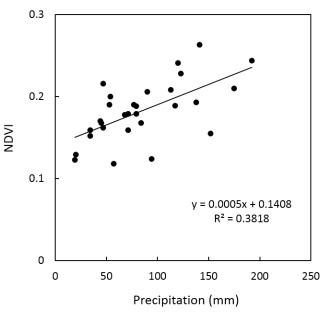

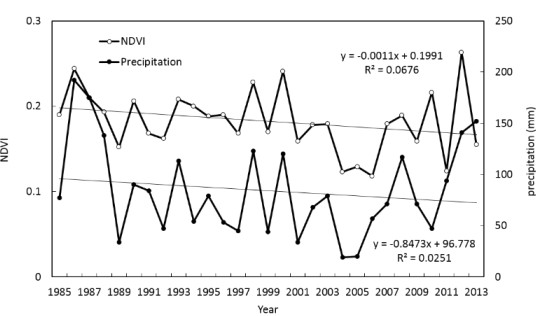

Figure 6 (a, b) Relationship between NDVI and precipitation during GS in Sainshand





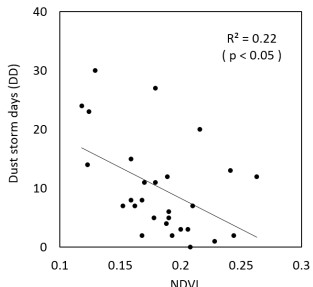
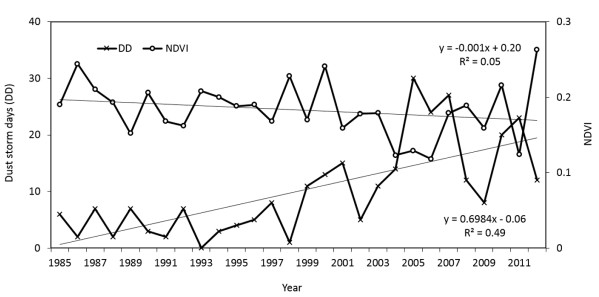

**Figure 7** (a, b) Relationship between dust storm days and NDVI in Sainshand

## Data availability

The NDVI 3g dataset used in this paper can be accessed and freely downloaded from the ECOCAST homepage (the GIMMS group at NASA's Goddard Space Flight Center):

https://ecocast.arc.nasa.gov/data/pub/gimms/.

Global Precipitation Climatology Project (GPCP) data can be downloaded from The NOAA/ESRL Physical Sciences Division (PSD) home page.

https://www.esrl.noaa.gov/psd/data/gridded/data.gpcp.html.

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
