# Peer review of "The Interactions Between Precipitation, Vegetation and Dust Emission Over Semi-Arid Mongolia"

_Atmospheric Chemistry and Physics, 2017_

## Short Comment (SC1) · 10 Mar 2017

Remote sensing technology applications in a large area for the vegetation survey is now a very advanced research methods. This research was focused in Gobi deserts with the precipitation and vegetation condition by using some vegetation indices. The global forest now experiencing serious problems with climate change. Large areas of forest death are due to insufficient water. In my research, i also found the same results with this article. Also, I think this article is very valuable. I give a high rating strongly recommended.

---

## Author Comment (AC1) · 10 Mar 2017

Thank you very much for your positive comment. In this study we just focus to semi-arid area vegetation how responsive to precipitation and how control to dust emissions.
* * *

---

## Short Comment (SC2) · 24 Mar 2017

This exciting paper is highly valuable in that it links remotely sensed vegetation information, climate data, and sand storm records both spatially and temporally. It takes full advantages of the temporal attribute of remote sensing (about 30 years' extent and find temporal resolution) and it offers insights of where and when to management vegetation coverage could be the most effective. There are some minor changes can be made to further improve the paper: 1) Change the extent of x-axis of fig 3a to 90-117.5 to match other figures; and 2) move some discussion about sandstorm occurrence in relation to vegetation coverage to result. In spite of these minor changes, I strongly recommend this paper.

---

## Short Comment (SC3) · 4 Apr 2017

In the recent time, arid and semi-arid part of Mongolia is frequently experienced droughts, mainly due to the increasing influence of the global warming. This is lead to the number of environmental changes, some are very serious not only for Mongolia, but also for other countries. In this paper, multi-annual data for the vegetation, precipitation and dust storm occurrences were analysed. The work is nicely illustrated and show important findings. I would like to recommend this paper for publication.

---

## Referee Comment (RC1) · Anonymous Referee #3 · 13 Jun 2017

The paper tried to clarify the interactions between precipitation, vegetation and dust emission over semi-arid Mongolia, however, as the authors said "the purpose of this study is to examine the relationship between the trend of vegetation variation and the effects of precipitation in the Gobi region". I believe that the authors give a sound analysis and fair results on the relationship between vegetation and precipitation.

However, there are few materials or data to be used to analyze the relationship between the precipitation and dust emission, and vegetation and dust emission, except for a single regression analysis "between summer vegetation and number of days in which dust storms occurred in the next spring in Sainshand city shown in Fig. 7a, b", although it is hard to be understand that the relationship was significant with such a low correlation coefficient, $R2 = 0.22$. Furthermore, it is also hard to be understand the

conclusion that "maintaining vegetation coverage during this period could reduce dust storm occurrence in the next spring" with the current single regression analysis with the data of only one site.

Based on my point of view, I suggest: 1) to focus on the relationship between vegetation and precipitation only and submit to other corresponded journals; or 2) to collect the more data of dust storm and analyze the relationship between the precipitation and dust emission, and vegetation and dust emission regionally. Anyway, I hope to rewrite and resubmit the paper with a native English check.

---

## Author Comment (AC3) · 14 Jun 2017

To Reviewer#3 We appreciate your very appropriate comment. First I'd like to answer your comment easily. Also, as you pointed out, I'd like to increase our data and make a corrections new version as soon as possible.

Reviewer's comment 1. There are few materials or data to be used to analyze the relationship between the precipitation and dust emission, and vegetation and dust emission, except for a single regression analysis "between summer vegetation and number of days in which dust storms occurred in the next spring in Sainshand city shown in Fig. 7a, b", although it is hard to be understand that the relationship was significant with such a low correlation coefficient, $R^2 = 0.22$. Authors answer: 1. The World Me-

teorological Observation Data (WMOD) that we used for this analysis included only 8 stations in our study area. Therefore, we extracted the data observed at Sainshand station as an example. However we can add at least seven results from analysis between NDVI and meteorological data observed at other seven stations.

Reviewer's comment 2. Furthermore, it is also hard to be understand the conclusion that "maintaining vegetation coverage during this period could reduce dust storm occurrence in the next spring" with the current single regression analysis with the data of only one site. Authors answer:

2. Vegetation in the semi-arid region is dominated by annual and perennial plants. Annual plants are highly dependent on rainfall and are also susceptible to overgrazing, while, perennial plants are relatively stable and can survive even in extreme drought conditions. However, once perennial plants such as shrub's die, they need a very long time to recover. For example, roots of many herbaceous plants can only extend 10 cm, they are very sensitive and response to rainfall but annual herbaceous plants are eaten by livestock, blown by the wind and lose until the following spring, so it's do not significantly affect the frequency of dust storms. Suaeda aralocaspica is a monoecious annual species commonly found in the Gobi desert and many perennial plants found in this region, especially shrubs are typified as Haloxylon ammodendron. However, annual plants do not influence so much for dust storm frequencies. On the other hands, perennial plants have very deep roots and this vegetation spots are effective in extracting water from their bare surroundings and therefore survive (Hardenberg et al., 2001), so they would not be affected by all precipitation. However, they can survive through winter to next spring and effect to frequency of outbreak of dust storm.

Please also note the supplement to this comment:
http://www.atmos-chem-phys-discuss.net/acp-2017-83/acp-2017-83-AC3-supplement.pdf

---

## Author Comment (AC4) · 15 Jun 2017

In the first manuscript of ACPD we used ground observation data of Sainshand weather station, but in this time we expanded the analysis area using WMO data. The World Meteorological Observation (WMO) data that we used for this analysis included only eight stations in our study area. See Fig. 7 (a,b,c,d,e,f,g,h)

Please also note the supplement to this comment:
http://www.atmos-chem-phys-discuss.net/acp-2017-83/acp-2017-83-AC4-supplement.pdf

[Figure]

2017.

Figure 7 (a, b, c, d, e, f, g, h) Relationship between dust storm days and growth season vegetation index (NDVI$_{GS}$) in Gobi region during 2000-2009 using the World Meteorological Observation (WMO) data

[Figure]

(a) Dashbalbar              (b) Saikhan-Ovoo              (c) Tsogt-Ovoo

**Fig. 1.**

**Supplement:**

Figure 7 (a, b, c, d, e, f, g, h) Relationship between dust storm days and growth season vegetation index (NDVI$_{GS}$) in Gobi region during 2000-2009 using the World Meteorological Observation (WMO) data

[Figure]

(a) Dashbalbar          (b) Saikhan-Ovoo          (c) Tsogt-Ovoo

Figure 7 (a, b, c, d, e, f, g, h) Relationship between dust storm days and growth season vegetation index (NDVI$_{GS}$) in Gobi region during 2000-2009 using the World Meteorological Observation (WMO) data

[Figure]

(d) Mandalgobi          (e) Bayandelger          (f) Sainshand

Figure 7 (a, b, c, d, e, f, g, h) Relationship between dust storm days and growth season vegetation index (NDVI $_{GS}$ ) in Gobi region during 2000-2009 using the World Meteorological Observation (WMO) data

[Figure]

(g) Zamin-Uud      (h) Dalanzadgad

---

## Author Comment (AC5) · 15 Jun 2017

In the first manuscript of ACPD we used ground observation data of Sainshand weather station, but in this time we expanded the analysis area using WMO data. The World Meteorological Observation (WMO) data that we used for this analysis included only eight stations in our study area. See Fig. 7 (a,b,c,d,e,f,g,h)

Please also note the supplement to this comment:
http://www.atmos-chem-phys-discuss.net/acp-2017-83/acp-2017-83-AC5-supplement.pdf

[Figure]

Figure 7 (a, b, c, d, e, f, g, h) Relationship between dust storm days and growth season vegetation index (NDVI $_{GS}$ ) in Gobi region during 2000-2009 using the World Meteorological Observation (WMO) data

[Figure]

(a) Dashbalbar        (b) Saikhan-Ovoo        (c) Tsogt-Ovoo

**Fig. 1.**

Figure 7 (a, b, c, d, e, f, g, h) Relationship between dust storm days and growth season vegetation index (NDVI $_{GS}$ ) in Gobi region during 2000-2009 using the World Meteorological Observation (WMO) data

[Figure]

(d) Mandalgobi        (e) Bayandelger        (f) Sainshand

**Fig. 2.**

Figure 7 (a, b, c, d, e, f, g, h) Relationship between dust storm days and growth season vegetation index (NDVI$_{GS}$) in Gobi region during 2000-2009 using the World Meteorological Observation (WMO) data

[Figure]

(g) Zamin-Uud          (h) Dalanzadgad

**Fig. 3.**

**Supplement:**

Figure 7 (a, b, c, d, e, f, g, h) Relationship between dust storm days and growth season vegetation index (NDVI$_{GS}$) in Gobi region during 2000-2009 using the World Meteorological Observation (WMO) data

[Figure]

(a) Dashbalbar          (b) Saikhan-Ovoo          (c) Tsogt-Ovoo

Figure 7 (a, b, c, d, e, f, g, h) Relationship between dust storm days and growth season vegetation index (NDVI$_{GS}$) in Gobi region during 2000-2009 using the World Meteorological Observation (WMO) data

[Figure]

(d) Mandalgobi          (e) Bayandelger          (f) Sainshand

Figure 7 (a, b, c, d, e, f, g, h) Relationship between dust storm days and growth season vegetation index (NDVI $_{GS}$ ) in Gobi region during 2000-2009 using the World Meteorological Observation (WMO) data

[Figure]

(g) Zamin-Uud      (h) Dalanzadgad

---

## Referee Comment (RC2) · Anonymous Referee #4 · 15 Jul 2017

This paper addresses the important issued of interaction between precipitation and vegetation in Mongolia. It uses the findings to suggest its impact on dust emissions and desertification. As presented data is predominantly NDVI and precipitation records at 4 sites the link between the two factors is examined. This suggests impact on ground cover; the connection to dust emission is assumed rather than documented. What is the dust data from Namhem? How can this be associated with vegetation and precipitation? The process is not clear. Mongolia is vast with varied landscape. At a 1o to 2.5 o scale is lost. More thorough investigation is needed such as groundtruthing, same-time location specific dust monitoring, or dust traps to make claims.

The paper can stress the NDVI-precipitation link which is useful. It highlights the landscape difference from east to west. Dust events have specific timeframes – this may

be different from the more general precipitation or vegetation. The connections are implied and logical, but the data is too general to say it is proved. It is not clear what dust data is used.

Similarly, the discussion on desertification is unspecific. The Gobi is dry with sparse vegetation that makes it a desert, not desertified. The question of processes that lead to degradation cannot be addressed with these research methods. It is mentioned that desertification is affected by human action in China. This is true with farming and development being major factors. This is not the case in Mongolia. It is more direct to focus on the correlational findings than to try to tie this to dust and desertification. More interpretation of correlations and differences would be useful. are they artefacts of the data, when/how data is collected? Why is there less coupling in the west? Is this because there is little precipitation and high variability? Plants are mentioned – are they important as cover, or for livestock – are plants palatable and is it implied that grazing is responsible for degradation? The research does not address these points directly so they should be presented as explanations only.

The paper will be strengthened by concentrating on the work that was done. Possible implications – dust, desertification, difference between Mongolia and China, possible climate change (please document) etc can go in a discussion. Factors that may affect vegetation, dust and desertification should also be mentioned. Is the implication that mining, herders, roads or agricultural is affecting land cover? Otherwise the data is in a vacuum – what is the significance, what do the findings mean in the real world. This does not have to be extensive, but without the context the purpose of the research is not clear. The title oversells the research so should more directly address the work done.

More specificity of terms, dates and timings would be useful as this can strengthen and clarify relationships. These details came across as general so were unconvincing. Why was only Shainshand used? Can the findings tell us more – some general results were what we would expect before undertaking the research.

The topic and engagement are good. A clearer sense of purpose would be useful. The authors can be encouraged to broaden the study or focus on a smaller area to provide more depth in the future. Also, consider some fieldwork to make the research less abstract. This would enable discussion of landscape, plants and degradation.

---

## Author Comment (AC7) · 19 Jul 2017

Thank you very much for your very valuable comment and question.

Question 1 What is the dust data from Namhem? How can this be associated with vegetation and precipitation? The process is not clear.

Answer 1: The "NAMHEM" is a part of the World Meteorological Organization (WMO) and it has 130 weather stations in the Mongolia. Dust storm data that we used is the number of dust storm occurrences in each day. However, we can use only this data observed at Sainshand station. RC1's Professor suggested that we should add more results of correlation analysis using dust storm data observed at another station. So we changed the data that we use from NAMHEM to the WMO SYNOP surface weather

data. This data was used as the number of dust storm days. This data is daily data and recorded as 1 when dust storms are observed visually in a day. We can use SYNOP data observed at 8 stations including Sainshand city in this study area. We used this data for single regression analysis to confirm the relationship between vegetation in GS season and number of days in which dust storms occurred in the following spring at each 8 stations. And we also confirmed the relationship between precipitation in same period as vegetation and number of dust storm days. We added the results from these analyses to our paper.

Question 2 Possible implications – dust, desertification, difference between Mongolia and China, possible climate change (please document) etc can go in a discussion. Factors that may affect vegetation, dust and desertification should also be mentioned. Is the implication that mining, herders, roads or agricultural is affecting land cover?

Answer 2: Previous studies suggested that the factors of desertification are almost same between Mongolia and China. Both of them are affected by climate factors and anthropogenic factors. However, socioeconomic factors were the dominant factor that affected desertification. For example, the cultivation, grazing, destruction or harvesting of herbaceous vegetation, and logging forests to produce firewood and rural construction materials in China (Feng et al., 2015). Furthermore, previous study showed that in addition to desertification, physical geographic conditions also influenced the frequency of dust occurrence (Wu et al., 2012). And also most of grassland are affected by human activity such as mining, road and especially grazing in Mongolia (Batjargal, 1997). Desertification is a kind of environment regime shift. However, the environment regime shifting was not found in this Mongolian sits (such as our study area) and it found in this Chinese sits (Sofue et al, 2017, Fig.1).

Question 1   What is the dust data from Namhem? How can this be associated with vegetation and precipitation? The process is not clear.

Answer 1:

The "NAMHEM" is a part of the World Meteorological Organization (WMO) and it has 130 weather stations in the Mongolia. Dust storm data that we used is the number of dust storm occurrences in each day. However, we can use only this data observed at Sainshand station. RC1's Professor suggested that we should add more results of correlation analysis using dust storm data observed at another station. So we changed the data that we use from NAMHEM to the WMO SYNOP surface weather data. This data was used as the number of dust storm days. This data is daily data and recorded as 1 when dust storms are observed visually in a day. We can use SYNOP data observed at 8 stations including Sainshand city in this study area.

We used this data for single regression analysis to confirm the relationship between vegetation in GS season and number of days in which dust storms occurred in the following spring at each 8 stations. And we also confirmed the relationship between precipitation in same period as vegetation and number of dust storm days. We added the results from these analyses to our paper.

Question 2  Possible implications – dust, desertification, difference between Mongolia and China, possible climate change (please document) etc can go in a discussion. Factors that may affect vegetation, dust and desertification should also be mentioned. Is the implication that mining, herders, roads or agricultural is affecting land cover?

Answer 2:

Previous studies suggested that the factors of desertification are almost same between Mongolia and China. Both of them are affected by climate factors and anthropogenic factors. However, socioeconomic factors were the dominant factor that affected desertification. For example, the cultivation, grazing, destruction or harvesting of herbaceous vegetation, and logging forests to produce firewood and rural construction materials in China (Feng et al., 2015). Furthermore, previous study showed that in addition to desertification, physical geographic conditions also influenced the frequency of dust occurrence (Wu et al., 2012). And also most of grassland are affected by human activity such as mining, road and especially grazing in Mongolia (Batjargal, 1997).

Desertification is a kind of environment regime shift. However, the environment regime shifting was not found in this Mongolian sits (such as our study area) and it found in this Chinese sits (Sofue et al, 2017, Fig.1).

**Fig. 1.**

---

## Author Comment (AC6)

**The Interactions Between Precipitation, Vegetation and Dust Emission Over Semi-Arid Mongolia**

Yuki Sofue[1], Buho Hoshino[2], Yuta Demura[3], Christopher McCarthy[4], Eunice Nduati[1], Akihiko Kondoh[1]

[1]Department of Earth Sciences, Chiba University, Chiba, 2638522, Japan
[2]Laboratory of Environmental Remote Sensing, Rakuno Gakuen University, Ebetsu, 0698501, Japan
[3] Field Researchers Corporation CO. Ltd, Sapporo, Hokkaido, 060-0007, Japan
[4] The Graduate School of Global Environmental Studies, Kyoto University, Japan

*Correspondence to*: Buho Hoshino (aosier@rakuno.ac.jp)

**Abstract.** Recently, droughts have become widespread in the Northern Hemisphere, including in Mongolia. The ground surface condition, particularly vegetation coverage affects the occurrence of dust storms. The main sources of dust storms in the Asian region are the Taklimakan and Gobi deserts. The purpose of this study is to clarify the relationship between precipitation and vegetation cover dynamics over 29 years in the Gobi region. We compared the patterns between precipitation and normalized difference vegetation index (NDVI) for a period of 29 years. The precipitation and vegetation datasets were examined to investigate the trends between 1985 to 2013. Cross correlation analysis between the precipitation and the NDVI anomalies was performed. Data analysis showed precipitation distribution patterns exhibited a decreasing trend with a shift from the east to the west of the study area. We found that vegetation in low precipitation areas was more degraded than that in high precipitation areas.

**1 Introduction**

Located in Central Asia, the Gobi includes a great desert and semi-arid region that stretches across huge portions of both Mongolia and China. The characteristic vegetation constitutes mixtures of grasslands, shrubs, saltwort and thorny trees. The Mongolian Gobi is a source for the formation of dust storms that sweep across East Asia (Natsagdorj et al., 2003). Dust storms frequently occur in arid and semi-arid regions and may have contributed to the desertification observed in recent decades as well as accelerated occurrence of more arid conditions over the drylands of Asia (Huang et al., 2014). Vegetation coverage is one of the most important factors for the reduction of dust storm occurrence (Ishizuka et al., 2005; Lee and Shon, 2009, 2011). Especially, it is known that spring dust frequency in China appears more correlated with NDVI from the prior summer than that in March to May of the same year (Zou and Zhai, 2004). Water is the main limiting factor for vegetation growth over southern Mongolia (Liu et al., 2013). However, both observation and modelling studies have indicated that an aridity trend is occurring and will occur most significantly in the semi-arid regions with droughts becoming more widespread in the Northern Hemisphere, including Asia, and particularly in Mongolia (e.g., Fu et al., 1999; Barlow et al., 2002; Dai et al., 1998; Lotsch et al., 2005). Furthermore, Huang et al. (2016) point out that the warming trends over drylands, particularly in arid regions, are

twice as great as those over humid regions. Sparsely vegetated drylands are an important source of dust emissions, but the mechanism of dust generation in response to timing of precipitation and the effects on soil and vegetation dynamics in these settings is still not well known (Urban et al., 2009). In this study, we used time series satellite vegetation measurements from the National Oceanic and Atmospheric Administration (NOAA) Advanced Very High Resolution Radiometer (AVHRR)

5   sensor to examine the variability and trends of land surface conditions in the Gobi region as represented by vegetation index data from 1985 to 2013.

**2 Data**

**2.1 NDVI data**

In this study, we use the Normalized Difference Vegetation Index (NDVI) to estimate vegetation variation. NDVI is given by

10   $$\text{NDVI} = \frac{NIR - RED}{NIR + RED} \tag{1}$$

Where, RED and NIR are the surface reflectance bands in the 550–700 nm (visible) and 730–1000 nm (infrared) regions of the electromagnetic spectrum, respectively. The NDVI3g data set used in this study is derived from measurements made by the AVHRR sensor aboard NOAA polar orbiting satellite series (NOAA-7, 9, 11, 14, 16). The NDVI3g data set is provided by the GIMMS group at NASA's Goddard Space Flight Center, as described by Tucker et al. (2005) and cover the period from

15   1981 to 2013, with a spatial resolution of 8km by 8km. The NDVI data were generated from processed 15-day NDVI composites using the maximum value compositing procedure to minimize effects of cloud contamination, varying solar zenith angles and surface topography (Holben, 1986). For this study, we subset the Gobi region covering the domain $90^0$ E - $117.5^0$ E and $40^0$ N - $47.5^0$ N, from the continental data set for the period from January 1985 to December 2013.

**2.2 Precipitation data**

20   The Global Precipitation Climatology Project (GPCP) data was derived from a joint analysis of satellite data and gauge data (Huffman et al., 2009) was used as precipitation data. This data has daily and monthly data. Daily data has $1^0 \times 1^0$ spatial resolution acquired between October 1996 and May 2015. And monthly data has $2.5^0 \times$ with a $2.5^0$ spatial resolution acquired between January 1979 and May 2015. Since, previous applications of NDVI in the Gobi region were focused mainly on the rainy season, NDVI patterns during the Growing Season (GS) were analyzed.

25   ### 2.3 Dust storm data

Data from NAMHEM (National Agency for Meteorology Hydrology and Environmental Monitoring, Mongolia) was used as the number of dust storm events. NAMHEM is part of the World Meteorological Organization (WMO) and has 130 weather stations in the Mongolia. The data used in this study was obtained from the Sainshand weather station (Station ID Number: 443540).

**3 Method**

We examined the spatiotemporal and seasonal variations, as well as the anomaly patterns for the monthly time series from 1985 to 2013. The growing season was defined by examining the long-term mean patterns of both precipitation and NDVI as shown in Fig. 1a and b, respectively, and with reference to long-term patterns of annual average precipitation distribution (Anyamba and Tucker, 2005). The months of May through September were selected to represent the average start and end of the Growing Season, referred to here as GS using Fig. 2(a, b). These figures were created from the results of calculating the monthly average of precipitation and NDVI during the study period. Fig. 1 shows the map of the average of all data for the study period from 1985 to 2013. This shows the long-term mean for this region. Interannual variability in the NDVI pattern was examined by calculating yearly GS anomalies as follows;

$$NDVI\sigma = \left[ (NDVI\alpha)/(NDVI\mu) - 1) \times 100 \right] \tag{2}$$

Where $NDVI\sigma$ are the respective GS percent anomalies, $NDVI\alpha$ are individual seasonal GS means and $NDVI\mu$ is the long-term GS mean. We also examined the precipitation anomalies during GS using the same method as that used for NDVI anomalies. Then we used the cumulative values of precipitation during GS. In addition, we performed cross correlation analysis between the cumulative precipitation and averaged NDVI for 15 days by averaging pixels in ($1°\times1°$) to verify the results of comparison with both trends. In this analysis, we use daily precipitation data and NDVI data resampled to ($1°\times1°$). The analysis period is determined by the period of daily precipitation data from 1996 to 2013.

**4 Results**

**4.1 Spatial patterns and trends**

The time series anomaly for the region are represented by the Hovmoller diagram for the period from January 1985 to December 2013 (Fig. 2a, 2b). It was considered that variation in vegetation arises due to a difference in conventional precipitation in the monsoon season. The amount of precipitation, which is supplied by monsoons from the Pacific and Indian Oceans, differs greatly between east and west. Following Fig. 3 (a, b), vegetation in the eastern region (from 110° E to 117.5° E) which had higher conventional precipitation had a higher response to the precipitation than the other regions. For example, in 1998 and 2012, it showed a higher than normal response to precipitation such as the period from 1990 to 1995. In the eastern region, low amounts of precipitation had been reported during 1999 to 2011. Nevertheless, vegetation anomaly was around 0 or more than 0. In contrast, in the central part of the study area ($100^0$ E from $110^0$ E), a high response of vegetation to higher precipitation was observed in only three years (1994, 1995, 2003). Also, there was no response to precipitation in 2012. The western region beginning from $100^0$ E to $90^0$ E showed a low response of vegetation to precipitation as compared to the east and central parts starting from 1985 due to the lower amount of conventional precipitation. Fluctuations in precipitation anomaly in the western region have increased since 2000. This is considered to be due to climate change and that the difference in the conventional precipitation determines the degree of influence. Also, it was found that the vegetation in the western part

of the study area is more vulnerable to climate change. The vegetation of the western region had a strong negative trend since 2010 and did not recover in the following years with greater values of precipitation. It is assumed that this decreasing trend might have promoted the further reduction of vegetation.

Time series of NDVI for selected locations across the Gobi region for the period from 1985 to 2013 are shown in Fig. 4. The data presented here are averaged NDVI values and cumulative precipitation for GS at each point. Sites 1 and 2 showed no change in trends of NDVI through the time series. On the other hand, site 3 showed a positive trend from 2003, and site 4 showed a negative trend from around 2009. There was a big difference in variation in NDVI values for sites 1 to 4. Sites 1 and 2 had a relatively large variation of NDVI year to year. Conversely sites 3 and 4 had a small variation. These variations depended on their response to precipitation.

**4.2 Cross correlation analysis**

The results of the cross-correlation analysis across the Gobi region for the period 1996 to 2013 are shown in Fig. 5 and the data is significant at $p < 0.05$ levels. The distribution of correlation coefficient is shown in Table 1. In the eastern region, there was a relatively high trend of correlation coefficient. By contrast, the time lag was larger and the correlation coefficient was very low in the western region, especially in the southwest area. The time lag was almost 0 and the vegetation conditions response within 15 days after precipitation in other regions. The positive relationship between NDVI and precipitation during GS in Sainshand is shown as an example (Fig. 6). The highest correlation value was 0.4 ($R^2 = 0.17$, $p < 0.05$) at time lag 0 locations. The vegetation had decreasing trends, but we postulate that it would recover in most locations during seasons with sufficient precipitation.

**5 Discussion and Conclusions**

Satellite measurements of vegetation dynamics in the Gobi region for a period of 29 years showed interannual variation and trends. In the Gobi region, precipitation is confined to the period from May to September. The variations of NDVI anomalies in the east region correspond well with the documented precipitation anomalies during this period. However, some parts, especially those in the southwest region of the Gobi region showed that the NDVI had decreased regardless of the precipitation amount.

In the arid and semi-arid Gobi region, vegetation cover is mainly constituted of annual and perennial plants. For example, Suaeda aralocaspica is a monoecious annual species commonly found in the Gobi desert and many perennial plants found in this region, especially shrubs are typified as Haloxylon ammodendron. Annual plants do not have a significant influence on dust storm frequencies directly. They exist as dead grass in the spring, but are not reflected in NDVI. However, rainfall encourages the growth of annually herbaceous plants and is recorded as a memory of biomass (Dry Matter Productivity), and in the following year they suppress dust emission as dry grass. The differences in dead grass coverage rates may increase or

decrease the outbreak of dust storms. The quantity of dust storm emissions tended to decrease along with an increased rate of the dead grass coverage areas with a maximum wind speed exceeded 9.1 m・s-1 in our study sites (Demura, et al., 2016). Similarly, there were some cases where the quantity of dust storm emissions had increased when the dead grass coverage areas had a decreased rate at the same maximum wind speed exceeded 9.1 m・s-1. In particular cases, the number of dust storm

5 emissions had a predilection to decrease along with an increased rate of the dead grass coverage even when the maximum wind speed exceeded 11 m・s-1 (Demura, et al., 2016). On the other hand, perennial plants have very deep roots and this type of vegetation are effective in extracting water from their bare surroundings and therefore survive (Hardenberg et al., 2001), so the affect of precipitation would be minimal. Furthermore, they can survive winter into the following spring and affect the frequency of dust storm outbreaks. However, once perennial plants, e.g. shrubs die, they need a substantial amount of time to

10 recover. This is a contributory factor to the occurrence of desertification.

Desertification can increase dust storms as has been observed in the Tibet Plateau and Hexi Corridor in recent years. This area is located in Northwest China, including the Tarim Basin. Perennial plants are dominant in this area due to low precipitation and desertification is therefore more likely to occur when there are drought conditions. This study focuses on the dynamic interaction between precipitation, vegetation (NDVI) and dust emission, however, only in the growing season (GS) are the

15 annual grasses reflected in NDVI.

Fig. 7a, b shows the relationship between summer vegetation and number of days in which dust storms occurred in the following spring at each meteorological station. These stations are located in the desert steppe zone of the study area. We analyzed this relationship using single regression analysis. From these results, the correlation coefficient were negative and the values were relatively high at Mandalgobi, Bayandelger, and Sainshand. These 3 places have more plant species than places

20 such as Tsogtovoo, Dalanzadgad and so on. Especially annual plant species are not stable and the amount can have an effect on the frequency of dust storm occurrence. Therefore, the conditions of vegetation coverage during GS might influence dust storm frequency. It has been suggested that maintaining vegetation coverage during this period could reduce dust storm occurrence in the following spring. Fig. 5 indicates that the vegetation condition in the southwest region of the Gobi should be monitored more carefully in the future.

25

**Acknowledgements**. This work was supported by JSPS KAKENHI Grant Numbers JP24340111,JP25550079 and the joint research of Rakuno Gakuen University. We are grateful to P. Tsedendamba and D. Munkhjargal for their support on field survey in Mongolia.

30

[Figure]

Figure 1: Long-term mean NDVI for the Gobi region (1985–2013) showing the transition from the eastern region with NDVI values of 0.6 to the west with values 0.02. The numbered locations indicate sites where NDVI data were extracted to examine the temporal variations and trends in NDVI from 1985 to 2013. Site 1 is located near Sainshand city, the capital of Dornogovi Province in Mongolia.

[Figure]

Figure 2: Hovmoller diagrams: (a) monthly precipitation and (b) monthly NDVI for the Gobi region averaged between $90^0$ E-$117.5^0$ E and $40^0$ N-$47.5^0$ N.

(a)
[Figure]
 (b)
[Figure]

Figure 3 Hovmoller diagrams: (a) precipitation anomaly and (b) NDVI anomaly for the Gobi region averaged between $90^0$ E-117.5$^0$ E and $40^0$ N-47.5$^0$N.

(a)
[Figure]
 (b)
[Figure]

10 Figure 4 Time series of average GS NDVI for selected sites across the Gobi region and cumulative GS precipitation during the same period (a, b). All sites, excluding site 4, exhibit a similar trend of precipitation variation over time (a). Site 4 shows minimal variation. On the one hand, the distribution patterns of NDVI have indicated a decreased amount of precipitation and a shift from the Northeast to Southwest region (b).

[Figure]

[Figure]

Figure 5 Map of the correlation between precipitation and NDVI, and distribution of time lag in response to precipitation.

Table 1 Correlation coefficient matrix.

| | 90°E | 91°E | 92°E | 93°E | 94°E | 95°E | 96°E | 97°E | 98°E | 99°E | 100°E | 101°E | 102°E | 103°E | 104°E | 105°E | 106°E | 107°E | 108°E | 109°E | 110°E | 111°E | 112°E | 113°E | 114°E | 115°E | 116°E | 117°E |
|---|---|---|---|---|---|---|---|---|---|---|---|---|---|---|---|---|---|---|---|---|---|---|---|---|---|---|---|---|
| 48°N | 0.47 | 0.56 | 0.55 | 0.51 | 0.43 | 0.53 | 0.56 | 0.65 | 0.67 | 0.68 | 0.71 | 0.70 | 0.72 | 0.66 | 0.66 | 0.61 | 0.65 | 0.65 | 0.66 | 0.66 | 0.63 | 0.65 | 0.65 | 0.64 | 0.66 | 0.61 | 0.64 | 0.61 |
| 47°N | **0.38** | 0.48 | 0.51 | 0.56 | 0.49 | 0.46 | 0.50 | 0.56 | 0.56 | 0.59 | 0.65 | 0.67 | 0.66 | 0.61 | 0.60 | 0.57 | 0.57 | 0.56 | 0.60 | 0.61 | 0.59 | 0.58 | 0.60 | 0.64 | 0.61 | 0.60 | 0.60 | 0.60 |
| 46°N | **0.29** | **0.34** | **0.40** | **0.48** | 0.48 | 0.43 | 0.47 | 0.50 | 0.44 | 0.48 | 0.52 | 0.58 | 0.50 | 0.46 | 0.50 | 0.51 | 0.47 | 0.45 | 0.46 | 0.48 | 0.55 | 0.58 | 0.62 | 0.63 | 0.64 | 0.63 | 0.60 | 0.61 |
| 45°N | **0.26** | **0.32** | **0.21** | **0.24** | **0.21** | 0.22 | **0.20** | 0.24 | 0.22 | 0.34 | 0.32 | 0.30 | 0.37 | 0.28 | 0.33 | 0.33 | 0.40 | 0.41 | 0.41 | 0.38 | 0.51 | 0.58 | 0.62 | 0.61 | 0.65 | 0.66 | 0.62 | 0.64 |
| 44°N | 0.37 | **0.24** | 0.40 | 0.51 | **0.43** | **0.19** | **0.17** | **0.18** | **0.17** | **0.16** | 0.27 | 0.28 | 0.34 | 0.42 | 0.43 | 0.44 | 0.40 | 0.36 | 0.42 | 0.46 | 0.52 | 0.56 | 0.60 | 0.61 | 0.66 | 0.68 | 0.68 | 0.68 |
| 43°N | **0.18** | **0.20** | **0.26** | **0.23** | **0.24** | **0.25** | **0.23** | **0.19** | **0.19** | **0.22** | **0.19** | 0.18 | 0.19 | 0.25 | 0.32 | 0.34 | 0.31 | 0.43 | 0.54 | 0.55 | 0.58 | 0.62 | 0.62 | 0.66 | 0.68 | 0.67 | 0.67 | 0.67 |
| 42°N | **0.17** | **0.17** | **0.18** | **0.19** | **0.15** | **0.23** | **0.25** | **0.23** | **0.19** | **0.19** | **0.23** | **0.19** | **0.21** | 0.20 | 0.24 | 0.31 | 0.48 | 0.56 | 0.62 | 0.63 | 0.65 | 0.65 | 0.69 | 0.67 | 0.69 | 0.67 | 0.65 | 0.66 |
| 41°N | **0.17** | **0.17** | **0.17** | **0.17** | **0.24** | **0.28** | **0.26** | 0.35 | **0.30** | 0.27 | **0.25** | **0.25** | 0.24 | 0.29 | 0.34 | 0.40 | 0.49 | 0.54 | 0.59 | 0.63 | 0.65 | 0.67 | 0.69 | 0.69 | 0.65 | 0.64 | 0.59 | 0.60 |

[Figure]

[Figure]

Figure 6 (a, b) Relationship between NDVI and precipitation during GS in Sainshand

[Figure]

**Figure 7** (a - h) Relationship between dust storm days and NDVI at each station

5   **Data availability**

The NDVI 3g dataset used in this paper can be accessed and freely downloaded from the ECOCAST homepage (the GIMMS group at NASA's Goddard Space Flight Center):

https://ecocast.arc.nasa.gov/data/pub/gimms/.

Global Precipitation Climatology Project (GPCP) data can be downloaded from The NOAA/ESRL Physical Sciences Division

10   (PSD) home page.

https://www.esrl.noaa.gov/psd/data/gridded/data.gpcp.html.

**References**

Anyamba, A., Tucker, C. J.: Analysis of Sahelian vegetation dynamics using NOAA-AVHRR NDVI data from 1981-2003, J ARID ENVIRON., 63, 596-614, 2005.

15   Barlow, M., Cullen, H., & Lyon, B.: Drought in central and southwest Asia: La Nina, the warm pool, and Indian Ocean precipitation. J CLIMATE, 15(7), 697-700, 2002.

Dai, A., Trenberth, K.E., Karl, T.R.: Global variations in droughts and wet spells: 1900–1995. GEOPHYS RES LETT 25, 3367–3370, 1998.

Dai, Aiguo., Trenberth, K. E., and Qian, T.: A Global Dataset of Palmer Drought Severity Index for 1870-2002: Relationship with Soil Moisture and Effects of Surface Warming, Journal of Hydrometeorology., 5, 1117-1130, 2004.

5  Demura Y., Hoshino B., Sofue Y., Kai K., Ts. Purevsuren, Baba K., Noda J.: Estimates of ground surface characteristics for outbreaks of the Asian Dust Storms in the sources region. ProScience, 3, 21-30, 2016. DOI:10.14644/dust.2016.004

Fu, C., Diaz, H. F., Dong, D., and Fletcher, J. O.: Changes in atmospheric circulation over Northern Hemisphere oceans associated with the rapid warming of the 1920s, INT J CLIMATOL., 19, 581-606, 1999.

Hardenberg, J., Meron, E., Shachak, M., and Zarmi, Y.: Diversity of vegetation patterns and desertification. PHYS REV LETT, 87(19), 198101, 2001.

10  Holben, B. N.: Characteristics of maximum-value composite images from temporal AVHRR data, INT J REMOTE SENS., 7, 1417-1434, 1986.

Huang, J., Wang, T., Wang, W., Li, Z. and Yan, H.: Climate effects of dust aerosols over East Asian arid and semiarid regions, J GEOPHYS RES : Atmospheres, 10.1002/2014JD021796, 2014.

15  Huang, J., Wang, T., Wang, W. Li, Z. and Yan, H.: Accelerated dryland expansion under climate change, Nature climate change, DOI: 10.1038/NCLIMATE2837 2016.

Huffman, G. J., Adler, R. F., Bolvin, D. T., Gu, G.: Improving the global precipitation record: GPCP Version 2.1, GEOPHYS RES LETT., 36, L17808, doi: 10.1029/2009GL040000, 2009.

Igarashi, Y., Inomata, Y., Aoyama, Y., Hirose, K., Takahashi, H., Shinoda, Y., Sugimoto, N., Shimizu, A., Chiba, M.: Possible

20  change in Asian dust source suggested by atmospheric anthropogenic radionuclides during the 2000s, ATMOS ENVIRON, 43, 2971-2980, 2009.

Ishizuka, M., Mikami, M. and Yamada, Y.: Anobservational study of soil moisture effects on wind erosion at a gobi site in the Taklimakan Desert, J GEOPHYS RES., 110, D18S03, doi:10.1029/2004JD004709, 2005.

Lee, E. H., Shon, B. J.: Examining the impact of wind and surface vegetation on the Asiandust occurrence over three classified

25  source regions, J GEOPHYS RES., 114, D06205, doi:10.1029/2008JD010687, 2009.

Lee, E. H., Shon, B. J.: Recent increasing trend in dust frequency over Mongolia and Inner Mongolia regions and its association with climate and surface condition change, ATMOS ENVIRON, 45, 4611-4616, 2011.

Liu, H., Tian, F., Hu, H.C., Hu, H.P. and Sivapalan, M.: Soil moisture controls on patterns of grass green-up in Inner Mongolia: an index based approach, HYDROL EARTH SYST SC., 17, 805-815, 2013.

30  Lotsch, A., Friedl, M.A., Anderson, B.T., Tucker, C.J. Response of terrestrial ecosystems to recent Northern Hemispheric drought. GEOPHYS RES LETT 32, L06705. doi:10.1029/2004GL022043, 2005.

Natsagdorj, L., Jugder, D., Chung, Y.S.: Analysis of dust storms observed in Mongolia during 1937-1999, ATMOS ENVIRON, 37, 1401-1411, 2003.

Tucker, C. J., Pinzon, J. E., Brown, M. E., Slayback, D. A., Pak, E. W., Mahoney, R., Vermote, E. F., Saleous, N. E.: An extended AVHRR 8‑km NDVI dataset compatible with MODIS and SPOT vegetation NDVI data, INT J REMOTE SENS., 26, 4485-4498, 2005.

Urban, F.E., Reynolds, R.L., Fulton, R.: The dynamic interaction of climate, vegetation, and dust emission, Mojave Desert,

5    USA. In: Fernandez-Bernal, A., De La Rosa, M.A. (Eds.), Arid Environments and Wind Erosion. NOVA Science Publishers, Inc., pp. 243-267, 2009.

Wang, X., Donga, Z., Zhang, J., Liu, L.: Modern dust storms in China: an overview, J ARID ENVIRON., 58, 559-574, 2004.

Yi. Y. Liu., J. P. Evans., M. F. McCabe., A. M. Richard., I. J. M. Albert van Dijk4, J. D. Albertus., S. Izuru.: Changing Climate and Overgrazing Are Decimating Mongolian Steppes, PLOS ONE., 8, 2, 2013.

10   Zou, X. K., and Zhai, P. M.: Relationship between vegetation coverage and spring dust storms over northern China, J GEOPHYS RES., 109(D3), 2004.